# Labeling of nascent RNA in the *C. elegans* intestine

**Omid Gholamalamdari**[1], **Stephanie C. Weber**[1,2]*

**1** Department of Biology, McGill University, Montreal, Canada, **2** Department of Physics, McGill University, Montreal, Canada

* steph.weber@mcgill.ca

## Abstract

Transcriptional regulation in *C. elegans* has been difficult to study at the level of nascent RNA because nucleotide analogs do not readily penetrate the cuticle. Here, we establish an ex vivo 5-ethynyl uridine (EU)-click labeling protocol that enables sensitive microscopy detection of newly transcribed RNA in dissected intestines. Using worms expressing fluorescent nucleolar markers, we show that EU incorporation faithfully reports on nascent transcription in both the nucleoplasm (mRNA) and the nucleolus (rRNA) and is abolished by inhibition of RNA polymerases. Spatial analysis further reveals that the majority of nascent rRNA transcripts localize to the fibrillar zone (FZ) of intestinal nucleoli, consistent with the conserved role of this compartment in rRNA synthesis. In addition to imaging applications, this workflow can be adapted for gene expression assays, providing a versatile approach for quantitative analysis of nascent transcription in *C. elegans*. By enabling direct visualization of nucleolar transcription in intact intestine tissue, this method opens new opportunities to investigate how nucleolar activity is regulated across development, aging, and disease contexts.

## Introduction

Transcription is the process that converts genetic information stored in the genome into RNA molecules that guide cellular function. Because transcription is dynamic and highly regulated, the ability to monitor RNA synthesis rate is critical for understanding how gene expression programs respond to developmental, environmental, and pathological cues.

Large-scale transcriptomic methods have been widely applied in the nematode *Caenorhabditis elegans* to measure steady-state RNA levels, providing important insights into gene regulation (reviewed in [1]). For example, single-cell RNA sequencing has been used to map transcriptional programs during embryogenesis [2], and tissue-specific RNA-seq approaches have profiled gene expression in adult neurons and the intestine [3,4]. However, RNA-seq mostly captures processed, stable

**Data availability statement:** Microscopy images, image processing pipelines and data analysis software can be found on borealis, The Canadian Dataverse Repository, at https://doi.org/10.5683/SP3/QUO9YR.

**Funding:** This work was funded by the Canadian Institutes of Health Research (PJT-159850 to SCW). This research was undertaken, in part, thanks to funding from the Canada Research Chairs Program (CRC-2020-00325 to SCW). The funders had no role in study design, data collection and analysis, decision to publish, or preparation of the manuscript. There was no additional external funding received for this study.

**Competing interests:** The authors have declared that no competing interests exist.

transcripts rather than the immediate products of transcription. To obtain spatial information, single-molecule fluorescence in situ hybridization (smFISH) has been applied to visualize specific RNAs in embryos and the gonad [2,5–8]. While FISH provides spatial information, it is limited to a handful of pre-selected targets and does not capture the dynamic nature of RNA synthesis.

A particularly powerful approach is to measure nascent RNA, which reflects the immediate transcriptional activity of RNA polymerases before RNA processing and turnover. Several methods have been developed to assay nascent transcription, including nuclear run-on assays, bromouridine (BrU) labeling, and 5-ethynyl uridine (EU) incorporation (reviewed in [9]). EU, in particular, has become widely adopted in various biological systems because of its membrane permeability [10–12]. Moreover, its alkyne group allows efficient and specific detection of labeled transcripts through copper-catalyzed click chemistry [13]. This approach enables rapid, sensitive, and minimally disruptive visualization of newly synthesized RNA.

Despite these advantages, measuring nascent transcription in multicellular organisms and intact tissues remains challenging. In many systems, the accessibility of nucleotide analogs is limited by cellular or extracellular barriers. In *C. elegans*, the cuticle surrounding the animal creates a permeability barrier preventing efficient uptake of EU [14], thereby hindering direct metabolic labeling of RNA in vivo. This has restricted the application of nascent RNA labeling in worms, even though *C. elegans* is a powerful model for studying the regulation of gene expression in development, stress, and aging.

Here, we establish a method to label nascent RNA in the *C. elegans* intestine by combining ex vivo dissection with EU incorporation and fluorescent detection via click chemistry. We validate that this approach reports on RNA polymerase transcription by demonstrating sensitivity to chemical and genetic perturbation of transcription. Using this workflow, we show that nascent rRNA localizes specifically to the fibrillar zone of intestinal nucleoli, consistent with the conserved role of this subcompartment in the early steps of ribosome biogenesis. Although currently optimized for microscopy, our approach can be readily adapted for molecular assays such as sequencing or RT-qPCR, providing a versatile tool for quantitative studies of nascent transcription in *C. elegans*.

## Methods

### Worm maintenance

Unless otherwise mentioned, strains were maintained at 20°C on NGM plates seeded with OP50 bacteria according to WormBook protocols [15]. Strains used in this work are detailed in S1 Table.

### EU-click

The protocol described in this peer-reviewed article is published on protocol.io, https://dx.doi.org/10.17504/protocols.io.rm7vzqr38vx1/v2 and is included for printing as S1 File with this article. Briefly, gravid adult worms (Day 1 of egg laying) were dissected in dissection media (60% v/v Libovitz's L-15 media, 20% Heat-inactivated FBS, 12.5 mM HEPES pH 7.5, Inulin 0.5 mg/mL, 0.04% w/v tetramisole) as described

in Laband et al. [16]. Dissected intestines were transferred to a silanized tube, and the dissection media was exchanged with RNA labeling media (60% v/v Libovitz's L-15 media, 20% Heat-inactivated FBS, 12.5 mM HEPES pH 7.5, Inulin 0.5 mg/mL, 1 mM EU). The intestines were then incubated for 5 minutes at room temperature (RT). The intestines were fixed with 60% 2-propanol for 10 minutes at RT. The intestines were then washed twice with PBS (KCl 2.7 mM, $KH_2PO_4$ 1.5 mM, NaCl 137 mM, $Na_2HPO_4$ 8.1 mM) + 0.1% Triton X-100 (PBS-Tx), permeabilized with PBS + 0.5% Triton X-100 for 5 minutes at RT, and then washed with PBS-Tx twice for 5 minutes at RT. Intestines were then incubated with click chemistry mixture (AlexaFluor594 0.08 mM, $CuSO_4$ 1.6 mM, THPTA 8 mM, vitamin C 3 mg/mL, in PBS-Tx) for 30 minutes at RT, washed twice with PBS-Tx for 5 minutes each. PBS-Tx was exchanged with mounting media (70% v/v glycerol, 0.03 mM Tris pH 9.0, 80 mg/mL n-propyl gallate, in nuclease-free $ddH_2O$) and samples were mounted on a slide before imaging.

### RNAi

RNAi was performed by feeding [17]. RNAi vectors were obtained from the Ahringer library and their sequences were verified. HT115 bacteria containing L4440 (control) and C36E8.1 (*tif-1a*) RNAi vectors were grown into log phase in LB + 50 µg/ml carbenicillin at 37°C, induced with 1 mM IPTG for 30 minutes at 37°C, and plated on RNAi plates (50 µg/ml Carb, 1 mM IPTG). Seeded plates were allowed to dry overnight at RT in the dark. Eggs from gravid worms were collected on control and *tif-1a* plates for 2 hours at 20°C and EU click was performed four days later (on Day 1 of adulthood).

### Actinomycin D and Flavopiridol mediated transcription inhibition

Dissected intestines were incubated in 45 µL of dissection media with DMSO (control), Actinomycin D (5 µg/mL), or Flavopiridol (5 µM). Intestines were incubated for 15 minutes (ActD) or 10 minutes (Flavopiridol) at RT before adding 5 µL of EU (10 mM). After 5 minutes of labeling at RT, intestines were fixed and click chemistry was performed.

### Imaging

Dissected intestines were imaged using a Leica DMI6000B inverted microscope, equipped with a Yokogawa CSU10 spinning disc confocal head, a Hamamatsu ImagEM EM-CCD camera and a 63X/1.40 NA Plan Apochromat objective lens. 491 nm and 561 nm lasers were used to excite nucleolar markers and EU-AF594, respectively.

### Image analysis

Custom software to measure the levels of nascent RNA and its localization within the nucleolus was developed in python (see Data Availability). Briefly, to define the nucleolar boundary, nucleolar protein markers were first segmented using "Workflow_npm1" from the Allen Cell Segmenter [18], followed by filling the holes in the segmented mask. To measure total nascent transcription, we summed the pixel intensities in the EU-AF594 channel within this nucleolar boundary. To calculate the degree of colocalization of nucleolar protein markers and nascent RNA, we used the Spearman correlation coefficient between the GFP (nucleolar protein) and EU-AF594 (nascent RNA) channels.

### Data analysis

All data processing and statistical analyses were performed in R. For comparison of two groups (3 batches each), significance was determined using two-way ANOVA.

## Results

### Labeling of nascent RNA in *C. elegans* intestine tissue

The *C. elegans* cuticle is largely impermeable to nucleotide analogs, and direct soaking of intact animals does not allow sufficient uptake of EU to detect nascent transcripts. To test whether increased cuticle permeability could improve labeling,

we soaked *dpy-9(e12)* and *dpy-10(e128)* mutant animals—known to have structurally compromised cuticles [14]—in 1 mM EU for 30 minutes. However, this EU pulse labeling followed by click chemistry did not produce detectable nascent RNA signal in these mutants.

We therefore developed an ex vivo approach inspired by methods using dissected intestines for gene expression analysis [19]. To maintain metabolic activity during dissection, handling, and labeling, we adapted a medium previously optimized for explanting and imaging of the *C. elegans* gonad [16]. Our procedure is illustrated in Fig 1. Briefly, worms are staged (Fig 1A), dissected under a dissecting scope to extrude the intestine (Fig 1B–1C), which are then transferred to tubes for labeling (Fig 1D–1E). Following fixation and click chemistry (Fig 1F), intestines are mounted and imaged (Fig 1G). This workflow can be applied across various life stages, including L4 larvae, young adults, and aged worms.

### EU-click procedure can quantitatively measure nascent transcription in int1 cells

To establish whether EU-click labeling can be used to measure nascent rRNA transcription in the intestine explants, we compared EU incorporation in control nucleoli and following inhibition of RNA polymerization. We employed worms

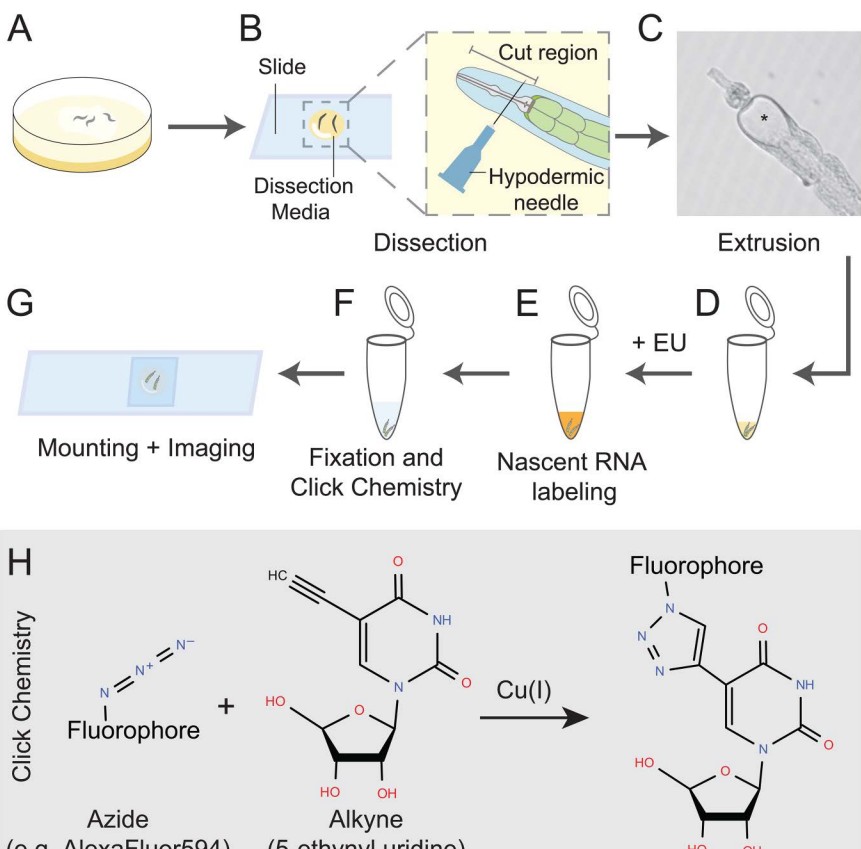

**Fig 1. Workflow for labeling nascent RNAs in the *C. elegans* intestine.** (A) Worms are grown on NGM plates to the desired life-stage (L4, adult, or aged animal). (B) Using a dissecting scope, animals are dissected in Dissection Media (DM). A hypodermic needle is used to make a single cut between the pharynx and the mouth, causing the intestine to extrude out of the worm body. (C) An example of an extruded intestine. Intestinal lumen is marked (*). (D) Dissected intestines are transferred to a tube. (E) DM is removed, and nascent RNA is labeled using RNA Labeling Media, followed by (F) fixation and click chemistry. (G) Intestines are mounted on a slide for imaging. (H) Click chemistry covalently combines the azide (fluorophore) and alkyne (EU).

expressing GARR-1::GFP from the endogenous locus. GARR-1, a component of the H/ACA snoRNP complex, localizes to the Fibrillar Zone (FZ) compartment of the nucleolus [20] and serves as a nucleolar marker.

Because the majority of transcripts generated within nucleoli are rRNAs, which are transcribed by RNA polymerase I, we first validated that the EU-AF594 signal indeed reports on nascent rRNA. Dissected intestines were treated with the RNA polymerase inhibitor actinomycin D (ActD). In control intestine tissue, the EU signal overlapped with GARR-1::GFP, consistent with incorporation into rRNA transcripts. ActD treatment substantially reduced EU incorporation in the nucleolus (Fig 2A).

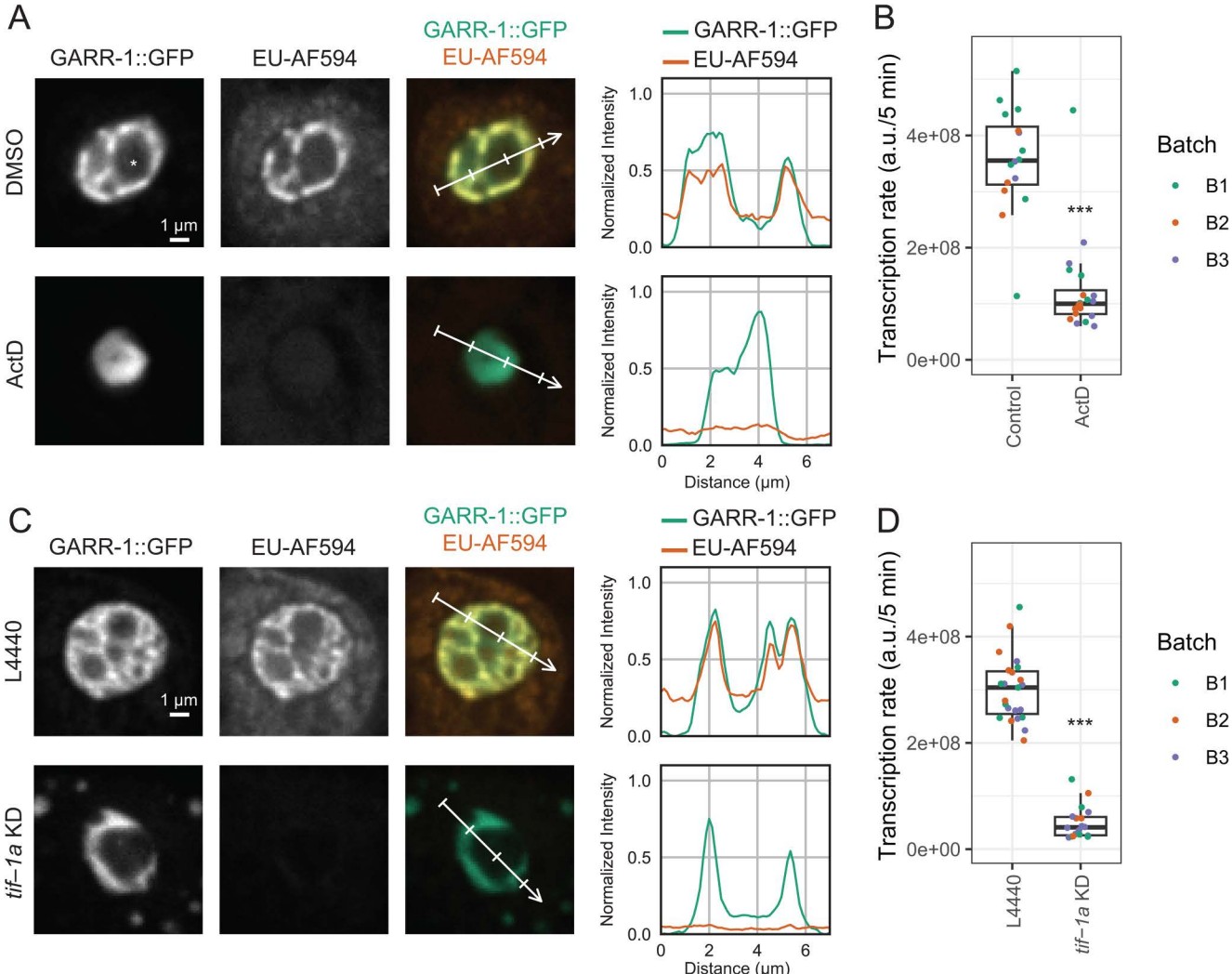

**Fig 2. EU-click can measure changes in nascent RNA transcription in the *C. elegans* intestine.** Images and quantitative measurements of nascent transcription (5 min pulse) with and without transcription inhibition. (A) Example nucleoli from int1 cells after treatment with DMSO (top) or ActD (bottom). Nucleolar vacuole is marked (*). Columns from left to right: GARR-1::GFP (FZ marker), EU conjugated with AF594 (EU-AF594), composite image of GARR-1::GFP (green) and EU-AF594 (orange), and intensity profiles of GFP and EU-AF594 signal along the line shown on the composite image. EU-AF594 pixel values in the grayscale and composite images and the line plots are normalized using the same range for DMSO and ActD. (B) Transcription rate (EU incorporated/5 min) in the nucleolus decreases upon transcription inhibition by ActD. Box plot comparing the sum of pixel values in the nucleolar mask for DMSO- and ActD-treated int1 cells. Points represent measurements from different worms; color indicates batch number. Significance was tested by two-way ANOVA using batch number and treatment as independent variables ($p < 1 \times 10^{-8}$). (C) Example nucleoli from int1 cells from L4440 (control) and *tif-1a* KD. Similar to A. (D) Transcription rate in nucleolus decreases upon *tif-1a* KD. Measurements and test of significance are similar to B ($p < 1 \times 10^{-14}$).

In control cells, nucleolar EU intensity was approximately 2.5-fold higher than nucleoplasmic levels (Fig 2A, top), whereas ActD treatment equalized nucleolar and nucleoplasmic signals (Fig 2A, bottom). We also noted that EU signal outside nucleoli was lower in ActD-treated cells, consistent with reduced transcription of mRNAs by RNA polymerase II, since ActD inhibits all RNA polymerases at the concentration used [21,22]. Quantification across biological replicates confirmed that ActD significantly decreased nucleolar EU levels compared to DMSO controls (two-way ANOVA, $p < 1 \times 10^{-8}$; Fig 2B).

We also detected an elevated EU signal in the nucleoplasm compared to the cytoplasm, consistent with transcription of nucleoplasmic class II RNAs (products of RNA polymerase II) (S1 Fig). We next asked if we could detect changes in nascent transcription of nucleoplasmic RNAs with our methodology. We used a highly specific inhibitor of RNA polymerase II, Flavopiridol (Flav) [21]. In Flav-treated intestines, the EU signal in the nucleoplasm was significantly lower compared to control. It is known that RNA pol II transcription of the Intergenic Sequence region of rDNA is essential for RNA pol I-mediated rRNA biogenesis in human cells [23], which explains the lower levels of nascent rRNA inside the nucleolus (S2 Fig). Moreover, this suggests that the nucleolar Pol-II-dependent mechanism that mediates rRNA synthesis is conserved in *C. elegans.*

To test EU-click labeling in a more physiological context, we next examined the effect of *tif-1a* gene knockdown (KD). *tif-1a* encodes a transcription factor required for RNA polymerase I activity [24]. GARR-1::GFP worms were fed bacteria expressing control (L4440) or *tif-1a* dsRNA from the egg stage to adulthood, followed by EU labeling. Knockdown of *tif-1a* led to a significant reduction in nucleolar EU signal relative to controls (two-way ANOVA, $p < 1 \times 10^{-14}$; Fig 2C–2D). In both ActD and *tif-1a* KD experiments, the reticulated FZ compartment collapses into a more compact structure, which is similar to formation of nucleolar caps upon transcription inhibition in human cells [25]. Together, these experiments demonstrate that EU-click labeling provides a sensitive, specific, and quantitative readout of nucleolar transcriptional activity in *C. elegans* intestinal cells.

### rRNA nascent transcripts localize to the fibrillar zone of int1 nucleoli

We next asked where within the nucleolus nascent rRNA transcription takes place. The *C. elegans* nucleolus has a bipartite organization [20,26], with an inner fibrillar zone (FZ), where rRNA transcription and early processing occur, and an outer granular zone (GZ), where late processing and ribosome assembly take place. In adult worms, intestinal nucleoli often contain a nucleolar vacuole which is devoid of FZ and GZ markers [26]. To distinguish between these compartments, we analyzed worm lines expressing GFP fusions to several nucleolar proteins: RPOA-2, GARR-1, and DAO-5, which localize to the FZ, and NUCL-1, which localizes to the GZ. Notably, the GARR-1 homolog marks the dense fibrillar component (DFC) of mammalian tripartite nucleoli [27].

We compared the spatial distribution of EU-labeled nascent RNA with each nucleolar marker. In int1 cells, the EU signal overlapped strongly with FZ markers (RPOA-2::GFP, GARR-1::GFP, and DAO-5::GFP), but not with the GZ marker NUCL-1::GFP (Fig 3A). Line-scan intensity profiles further demonstrated that EU incorporation peaks coincide with the FZ signal, whereas little overlap was observed with the GZ signal (Fig 3A–right). To quantitatively compare the location of transcription relative to protein markers, we calculated the Spearman correlation coefficients. As expected, the Spearman correlation between EU and GFP signals were significantly higher for RPOA-2, a subunit of RNA polymerase I, among FZ markers and lowest for NUCL-1 (Fig 3B).

Together, these results show that nascent rRNA transcription in the *C. elegans* intestinal nucleolus localizes specifically to the FZ compartment, consistent with the conserved role of this subdomain in rRNA synthesis (Fig 3C).

### Discussion

By combining intestinal dissection with EU incorporation and click chemistry, we overcame the barrier of cuticle impermeability that has previously prevented metabolic RNA labeling in *C. elegans*. Our validation experiments demonstrate that the EU signal reflects RNA transcription, as it is strongly reduced by actinomycin D treatment. Knockdown of *tif-1a*

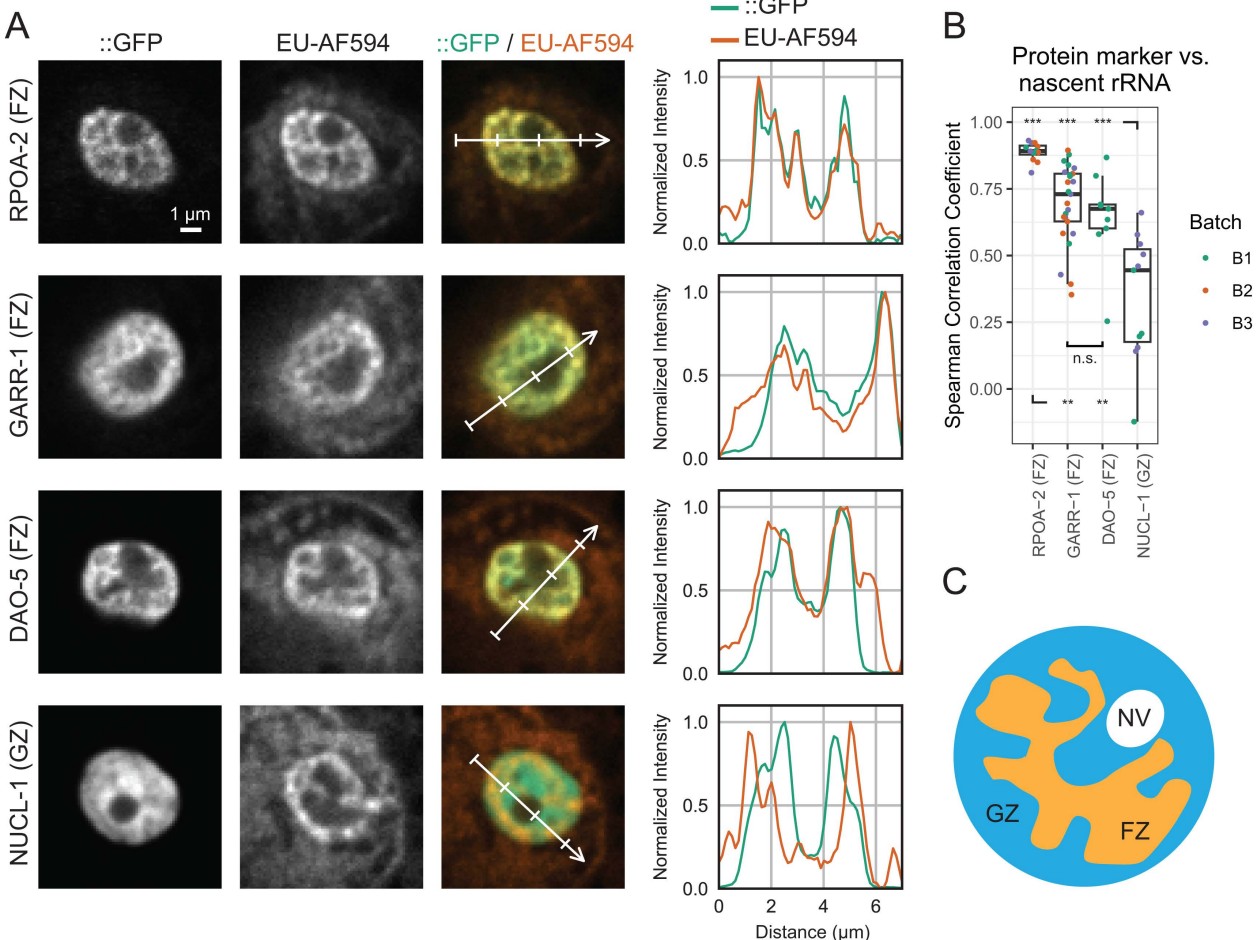

**Fig 3. Nascent RNA transcription overlaps with the FZ compartment of nucleoli.** (A) Example nucleoli of int1 cells from worm lines expressing different GFP labeled protein markers localized to Fibrillar Zone (FZ) or Granular Zone (GZ) compartments. Top to bottom: RPOA-2::GFP (FZ), GARR-1::GFP (FZ), DAO-5::GFP (FZ), and NUCL-1::GFP (GZ). Columns from left to right: nucleolar protein::GFP (::GFP), EU-AF594, composite image of ::GFP (green) and EU-AF594 (orange), and intensity profiles of GFP and AF594 signal along the line shown on the composite image. GFP and EU-AF594 pixel values in the line plots are normalized (min/max) along each line. (B) Box plots comparing the Spearman correlation of pixel values between GFP (nucleolar protein) and EU-AF594 (nascent RNA). Points represent measurements from different worms. Significance was tested using two-way ANOVA and Tukey-Kramer test. P-values for each pairwise comparison: not significant (n. s.) for DAO-5 vs. GARR-1, $p < 0.001$ (***) for NUCL-1 vs. RPOA-2, DAO-5, and GARR-1, and $p < 0.01$ (**) for RPOA-2 vs. DAO-5 and GARR-1. (C) Nascent rRNA transcription localizes to the FZ compartment of the nucleolus in the *C. elegans* intestine. GZ: Granular Zone; FZ: Fibrillar Zone; NV: Nucleolar Vacuole.

(transcription factor of RNA polymerase I) significantly reduced nascent RNA signal and revealed that the majority of nascent transcription is rRNA (product of RNA polymerase I) and localized to the nucleolus. We employed this for spatial analysis of the nucleolus and demonstrated that nascent rRNA localizes specifically to the fibrillar zone of intestinal nucleoli, consistent with the conserved role of this subcompartment in rRNA synthesis [28]. Together, these results establish a robust and versatile method for visualizing and quantifying nascent transcription in intact *C. elegans* tissues.

Standard EU-click chemistry has been widely applied in cultured cells [11–13,29], but extending such labeling to whole organisms often requires organism-specific adaptations. In bacteria, for example, EU and 4sU incorporation suppress cell growth, so alternative analogs must be used [30]. In *Drosophila*, 4-thiouridine (4-TU) can be delivered to larvae by feeding and combined with tissue-specific expression of uracil phosphoribosyltransferase (UPRT) to achieve nascent transcript

labeling [31,32]. In vertebrate systems, nucleotide analogs have been directly injected into zebrafish embryos [33] and mouse pups [34]. These examples highlight the technical innovations necessary for metabolic RNA labeling across different model systems. Notably, *C. elegans* is absent from recent surveys of such approaches [35], underscoring the novelty of our method as the first to achieve reliable metabolic labeling of nascent RNA in worms.

Beyond providing a methodological advance, our approach yields new insights into nucleolar organization in *C. elegans*. By combining EU labeling with compartment-specific markers, we find that the EU signal localizes to a reticulated network within the fibrillar zone, contrasting with the discrete transcriptional foci often reported in cultured mammalian cells [36,37]. This observation suggests that the nucleolus of *C. elegans* may execute transcription and early processing in a structurally distinct manner compared to the nucleoli of higher eukaryotes. Moreover, DAO-5 and GARR-1 localize within *C. elegans* nucleoli beyond sites of nascent rRNA transcription, whereas RPOA-2 colocalizes primarily with nascent rRNA. Together, these patterns challenge a strictly bipartite nucleolar organization [36] and suggest the presence of a DFC-like compartment in worms. Whether this reflects evolutionary differences in nucleolar substructure or organism-specific adaptations remains an open question, but our method provides a foundation for exploring such structural–functional relationships in vivo.

## Supporting information

**S1 File. Detailed nascent RNA labeling protocol.**
(PDF)

**S1 Table. List of *C. elegans* strains used in this study.**
(DOCX)

**S1 Fig. EU signal is also detected in the nucleoplasm which may correspond to the activity of RNA polymerase II.** A low magnification image of int1 cell. Columns from left to right: NUCL-1::GFP, EU conjugated with AF594 (EU-AF594), composite image of NUCL-1::GFP (green) and EU-AF594 (orange), and intensity profiles of GFP and EU-AF594 signal along the line shown on the composite image. The nucleus and nucleolus boundaries are marked with long- and short-dashed lines, respectively.
(EPS)

**S2 Fig. Nascent RNA labeling protocol can detect nascent transcript changes in the nucleoplasm outside of the nucleolus.** Images and quantitative measurements of nascent transcription (5 min pulse) with and without Flavopiridol (Flav)-mediated RNA Pol II inhibition. (A) Example nucleoli from int1 cells after treatment with DMSO (top) or Flav (bottom). Columns from left to right: NUCL-1::GFP (GC marker), EU conjugated with AF594 (EU-AF594), composite image of NUCL-1::GFP (green) and EU-AF594 (orange), and intensity profiles of GFP and EU-AF594 signal along the line shown on the composite image. EU-AF594 pixel values in the grayscale and composite images and the line plots are normalized using the same range for DMSO and Flav. The nucleolus and expanded nucleolus boundaries (used for nucleoplasmic EU concentration measurements) are marked with short- and long-dashed lines, respectively. (B) The concentration of nascent RNA in the nucleoplasm (B) and nucleolus (C) decreases upon transcription inhibition by Flav. Box plot comparing the EU concentration for DMSO- and Flav-treated int1 cells. Points represent measurements from different worms; color indicates batch number. Significance was tested by two-way ANOVA using batch number and treatment as independent variables (***: $p < 1 \times 10^{-3}$).
(EPS)

## Acknowledgments

We thank Drs. Laeya Baldini, Eric Cheng, Abigail Gerhold, and Réda Zellag for technical and scientific suggestions that led to the optimization of this protocol and manuscript.

## Author contributions

**Conceptualization:** Omid Gholamalamdari, Stephanie C. Weber.

**Data curation:** Omid Gholamalamdari.

**Formal analysis:** Omid Gholamalamdari.

**Funding acquisition:** Stephanie C. Weber.

**Investigation:** Omid Gholamalamdari.

**Methodology:** Omid Gholamalamdari.

**Project administration:** Stephanie C. Weber.

**Software:** Omid Gholamalamdari.

**Supervision:** Stephanie C. Weber.

**Visualization:** Omid Gholamalamdari.

**Writing – original draft:** Omid Gholamalamdari.

**Writing – review & editing:** Stephanie C. Weber.

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
