## [Decision Letter · Decision Letter 0]

25 Oct 2025

PONE-D-25-50225Labeling of Nascent RNA in the C. elegans IntestinePLOS ONE

Dear Dr. Weber,

Thank you for submitting your manuscript to PLOS ONE. After careful consideration, we feel that it has merit but does not fully meet PLOS ONE’s publication criteria as it currently stands. Therefore, we invite you to submit a revised version of the manuscript that addresses the points raised during the review process.

Your manuscript was reviewed by two experts. As you can see from the reviews, there was general interest but the reviewers commented on technical issues and clarity. Please respond to all of the reviewers' comments. I believe addressing these comments will improve the manuscript.==============================

We look forward to receiving your revised manuscript.

Kind regards,

Eric Jan, Ph.D.

Academic Editor

PLOS ONE

Journal Requirements:

This work was funded by a Canadian Institutes of Health Research Project grant (PJT-159850) awarded to SCW. This research was undertaken, in part, thanks to funding from the Canada Research Chairs Program to SCW.

This work was funded by a Canadian Institutes of Health Research Project grant (PJT-159850) awarded to SCW. This research was undertaken, in part, thanks to funding from the Canada Research Chairs Program to SCW.

5. Thank you for uploading your study's underlying data set. Unfortunately, the repository you have noted in your Data Availability statement does not qualify as an acceptable data repository according to PLOS's standards.

7. We note that you have included the phrase “data not shown” in your manuscript. Unfortunately, this does not meet our data sharing requirements. PLOS does not permit references to inaccessible data. We require that authors provide all relevant data within the paper, Supporting Information files, or in an acceptable, public repository. Please add a citation to support this phrase or upload the data that corresponds with these findings to a stable repository (such as Figshare or Dryad) and provide and URLs, DOIs, or accession numbers that may be used to access these data. Or, if the data are not a core part of the research being presented in your study, we ask that you remove the phrase that refers to these data.

Reviewers' comments:

Reviewer's Responses to Questions

**Comments to the Author**

1. Does the manuscript report a protocol which is of utility to the research community and adds value to the published literature?

Reviewer #1: Yes

Reviewer #2: Yes

2. Has the protocol been described in sufficient detail?

To answer this question, please click the link to protocols.io in the Materials and Methods section of the manuscript (if a link has been provided) or consult the step-by-step protocol in the Supporting Information files.

The step-by-step protocol should contain sufficient detail for another researcher to be able to reproduce all experiments and analyses.

Reviewer #1: Yes

Reviewer #2: Yes

3. Does the protocol describe a validated method?

Reviewer #1: Yes

Reviewer #2: Yes

4. If the manuscript contains new data, have the authors made this data fully available?

Reviewer #1: Yes

Reviewer #2: Yes

**5. Is the article presented in an intelligible fashion and written in standard English?**

Reviewer #1: Yes

Reviewer #2: Yes

6. Review Comments to the Author

Reviewer #1: This paper reports a new method for detecting nascent ribosomal RNA (rRNA) in C. elegans intestinal cells. To overcome the impermeability of the C. elegans cuticle, the intestine is removed by dissection for incubation with EU, a uridine derivative that can be labeled by click chemistry for quantitative light microscopy. The method is clearly described and accompanied by a detailed protocol posted at protocols.io. Quantitative imaging detects a robust EU signal in the nucleolus, the established site or rRNA synthesis. RNAi of a canonical transcription factor for rRNA expression depletes the nucleolar EU signal lending strong support to the hypothesis that this method reliably detects nascent rRNA. Lastly, live cell markers (GFP) for nucleolar compartments appear to confirm that rRNA is largely localized to the fibrillar zone (FZ) (see below). This manuscript offers a useful and well-validated protocol for measuring rRNA in the C. elegans intestine and could be potentially useful for detection of nascent RNA in other C. elegans dissected tissues or isolated cells.

Minor Revisions:

1. A key finding is that rRNA transcripts are localized to the nucleolar fibrillar zone (FZ). Figure 3B quantifies co-localization (Spearman correlation coefficient) of the EU-594 signal with FZ markers DOA-1 and GARR-1 vs the granular zone (GZ) marker NUCL-1. Student’s t-test shows no significant difference (ns) between DOA-1 and GARR-1 but no statistics are reported for a comparison of NUCL-1 vs DOA-1 and GARR-1. This statistical test is important for substantiating the authors’ conclusion that the “majority of nascent rRNA transcripts localize to the fibrillar zone.“

2. Because intestinal cell nuclei are polyploid, it should be possible to detect other classes of nascent transcripts. This finding would provide a useful demonstration of the broad applicability of the protocol.

3. Need reference for concentration of ActD sufficient to inhibit RNA Pol II (line 154)

4. A figure illustrating click chemistry would be useful to readers unfamiliar with this method.

Reviewer #2: In this manuscript, Gholamalamdari and Weber present a protocol for nascent labelling of RNA in dissected intestines in C. elegans. Historically, the worm cuticle has prevented facile labelling of RNA with nucleoside analogs. Here, the authors solve this problem by extrusion of the intestine, allowing efficient 5-ethynyl uridine (EU) incorporation. The authors provide good evidence for nascent transcript labelling in the nucleolus, demonstrating reduced incorporation in response to perturbations of transcription, such as treatment with actinomycin D or knockdown of tif-1a (RNA polymerase I transcription factor). Subsequently, the authors further use this technique to explore the location of rRNA transcription within the nucleolus, by co-staining nascent RNA with known markers of the Fibrillar Zone (FZ) and Granular Zone (GZ). Their results demonstrate that the EU signal localizes within the FZ.

Overall, I enjoyed reading this protocol and I think it will be a good addition for the C. elegans community interested in studying gene regulation and transcription.

I only have minor comments that would be helpful to aid others interested in trying this technique.

1) It would be good for the authors to provide some rough guidelines/recommendations on how many animals should be dissected for a typical imaging experiment. It’s not clear how much material loss takes place after the manipulations.

2) Do the authors have some lower magnification images of the labelled intestines? Does the nucleolar/rRNA signal make it difficult to observe mRNA in the nucleus or cytoplasm? This would be interesting to know for anyone interested in studying other RNA species in the cell.

7. PLOS authors have the option to publish the peer review history of their article (what does this mean?). If published, this will include your full peer review and any attached files.

Reviewer #1: No

Reviewer #2: No

---

## [Author Response · Author response to Decision Letter 1]

5 Jan 2026

We thank the reviewers for their thoughtful and constructive feedback. Based on their helpful suggestions, we have revised our manuscript to include new experimental data, as well as clarifications to the text and figures. Below, we provide a point-by-point response to accompany the revised manuscript.

Reviewers' comments:

Reviewer's Responses to Questions

Comments to the Author

1. Does the manuscript report a protocol which is of utility to the research community and adds value to the published literature?

Reviewer #1: Yes

Reviewer #2: Yes

2. Has the protocol been described in sufficient detail?

To answer this question, please click the link to protocols.io in the Materials and Methods section of the manuscript (if a link has been provided) or consult the step-by-step protocol in the Supporting Information files.

The step-by-step protocol should contain sufficient detail for another researcher to be able to reproduce all experiments and analyses.

Reviewer #1: Yes

Reviewer #2: Yes

3. Does the protocol describe a validated method?

Reviewer #1: Yes

Reviewer #2: Yes

4. If the manuscript contains new data, have the authors made this data fully available?

Reviewer #1: Yes

Reviewer #2: Yes

5. Is the article presented in an intelligible fashion and written in standard English?

Reviewer #1: Yes

Reviewer #2: Yes

6. Review Comments to the Author

Reviewer #1: This paper reports a new method for detecting nascent ribosomal RNA (rRNA) in C. elegans intestinal cells. To overcome the impermeability of the C. elegans cuticle, the intestine is removed by dissection for incubation with EU, a uridine derivative that can be labeled by click chemistry for quantitative light microscopy. The method is clearly described and accompanied by a detailed protocol posted at protocols.io. Quantitative imaging detects a robust EU signal in the nucleolus, the established site or rRNA synthesis. RNAi of a canonical transcription factor for rRNA expression depletes the nucleolar EU signal lending strong support to the hypothesis that this method reliably detects nascent rRNA. Lastly, live cell markers (GFP) for nucleolar compartments appear to confirm that rRNA is largely localized to the fibrillar zone (FZ) (see below). This manuscript offers a useful and well-validated protocol for measuring rRNA in the C. elegans intestine and could be potentially useful for detection of nascent RNA in other C. elegans dissected tissues or isolated cells.

Minor Revisions:

1. A key finding is that rRNA transcripts are localized to the nucleolar fibrillar zone (FZ). Figure 3B quantifies co-localization (Spearman correlation coefficient) of the EU-594 signal with FZ markers DOA-1 and GARR-1 vs the granular zone (GZ) marker NUCL-1. Student’s t-test shows no significant difference (ns) between DOA-1 and GARR-1 but no statistics are reported for a comparison of NUCL-1 vs DOA-1 and GARR-1. This statistical test is important for substantiating the authors’ conclusion that the “majority of nascent rRNA transcripts localize to the fibrillar zone.“

We performed two-way ANOVA and Tukey-Kramer test on these data and now include p-values for each pairwise comparison (Fig 3). Consistent with our original conclusion, there is no statistical difference (p > 0.05) between the Spearman correlation coefficients for EU signal with DAO-5 and GARR-1. We now explicitly report statistical differences for all other marker pairs: p < 0.001 for NUCL-1 vs. RPOA-2, DAO-5, and GARR-1, and p < 0.01 for RPOA-2 vs. DAO-5 and GARR-1. These results further validate the distinct localization patterns of our FZ (RPOA-2, DAO-5, GARR-1) and GZ (NUCL-1) markers, and support the conclusion that the majority of EU signal localizes to the FZ.

2. Because intestinal cell nuclei are polyploid, it should be possible to detect other classes of nascent transcripts. This finding would provide a useful demonstration of the broad applicability of the protocol.

The reviewer raises an excellent question. To test whether we could detect other classes of nascent RNA, we performed additional experiments in the presence of flavopiridol, a fast-acting inhibitor of RNA polymerase II. Unexpectedly, we saw a significant reduction in EU signal not only in the nucleoplasm, but also in the nucleolus itself. We suspect that this is due to Pol II’s role in transcribing noncoding RNA from the intergenic spacers between rDNA repeats (PMID: 32669707). We have added these results as a supplementary figure (S2 Fig).

3. Need reference for concentration of ActD sufficient to inhibit RNA Pol II (line 154)

We have added two references to support this statement (line 183). The first (PMID: 21922053) gives concentration guidelines for inhibition of Pol I and Pol II primarily in the context of cultured cells. The second (PMID: 38293118) provides an example of actinomycin D used to inhibit Pol II transcription in C. elegans.

4. A figure illustrating click chemistry would be useful to readers unfamiliar with this method.

We thank the reviewer for this suggestion. We have added a panel (H) to Figure 1, illustrating the chemical structure of 5-EU and the click reaction.

Reviewer #2: In this manuscript, Gholamalamdari and Weber present a protocol for nascent labelling of RNA in dissected intestines in C. elegans. Historically, the worm cuticle has prevented facile labelling of RNA with nucleoside analogs. Here, the authors solve this problem by extrusion of the intestine, allowing efficient 5-ethynyl uridine (EU) incorporation. The authors provide good evidence for nascent transcript labelling in the nucleolus, demonstrating reduced incorporation in response to perturbations of transcription, such as treatment with actinomycin D or knockdown of tif-1a (RNA polymerase I transcription factor). Subsequently, the authors further use this technique to explore the location of rRNA transcription within the nucleolus, by co-staining nascent RNA with known markers of the Fibrillar Zone (FZ) and Granular Zone (GZ). Their results demonstrate that the EU signal localizes within the FZ.

Overall, I enjoyed reading this protocol and I think it will be a good addition for the C. elegans community interested in studying gene regulation and transcription.

I only have minor comments that would be helpful to aid others interested in trying this technique.

1) It would be good for the authors to provide some rough guidelines/recommendations on how many animals should be dissected for a typical imaging experiment. It’s not clear how much material loss takes place after the manipulations.

We have added the following recommendation to the guideline section of the protocol:

For each biological or technical replicate we recommend dissecting 20 worms. Due to the material loss during the procedure the recovery rate is between 5 to 10 worms for imaging.

2) Do the authors have some lower magnification images of the labelled intestines? Does the nucleolar/rRNA signal make it difficult to observe mRNA in the nucleus or cytoplasm? This would be interesting to know for anyone interested in studying other RNA species in the cell.

We have added lower-magnification images to Figure S1, and provide line profiles to highlight the differences in EU signal among the nucleolus, nucleoplasm, and cytoplasm.

Since Reviewer #1 asked a similar question, we performed additional experiments in the presence of flavopiridol, a fast-acting inhibitor of RNA polymerase II. Unexpectedly, we saw a significant reduction in EU signal not only in the nucleoplasm, but also in the nucleolus itself. We suspect that this is due to Pol II’s role in transcribing noncoding RNA from the intergenic spacers between rDNA repeats (PMID: 32669707). We have added these results as a supplementary figure (S2 Fig).

7. PLOS authors have the option to publish the peer review history of their article (what does this mean?). If published, this will include your full peer review and any attached files.

Do you want your identity to be public for this peer review? For information about this choice, including consent withdrawal, please see our Privacy Policy.

Reviewer #1: No

Reviewer #2: No

---

## [Editor Report · Decision Letter 1]

11 Jan 2026

Labeling of nascent RNA in the C. elegans intestine

PONE-D-25-50225R1

Dear Dr. Weber,

We’re pleased to inform you that your manuscript has been judged scientifically suitable for publication and will be formally accepted for publication once it meets all outstanding technical requirements.

Kind regards,

Eric Jan, Ph.D.

Academic Editor

PLOS One
---

## [Editor Report · Acceptance letter]

PONE-D-25-50225R1

PLOS One

Dear Dr. Weber,

I'm pleased to inform you that your manuscript has been deemed suitable for publication in PLOS One. Congratulations! Your manuscript is now being handed over to our production team.

Kind regards,

on behalf of

Dr. Eric Jan

Academic Editor

PLOS One